# Adoption of policies to improve respectful maternity care in Timor-Leste

**Angelina da Costa Fernandes**[1]*, **Stefanus Supriyanto**[2], **Chatarina Umbul Wahyuni**[2], **Hari Basuki Notobroto**[2], **Alexandra Gregory**[3], **Kayli Wild**[3]

1 Doctoral Program of Public Health, Airlangga University, Surabaya, Indonesia, 2 School of Public Health, Airlangga University, Surabaya, Indonesia, 3 Centre for Child Development and Education, Menzies School of Health Research, Charles Darwin University, Northern Territory, Australia

* angelina.da.costa-2019@fkm.unair.ac.id

**Data Availability Statement:** All relevant data are within the manuscript and its Supporting information files.

## Abstract

### Introduction

There are now well-established global standards for supporting improvement in women's experience of maternity services, including frameworks for the prevention of mistreatment during childbirth. To support initiatives to improve the quality of care in maternal health services in Timor-Leste, we examine the adoption of global respectful maternity care standards in the national intrapartum care policy and in three urban birth facilities in Dili.

### Methods

From May to July 2022, we conducted a desk review of the Timor-Leste National Intrapartum Care Standards and Clinical Protocols for Referral Facilities and Community Health Centres. This was followed by a health-facility audit of policies, guidelines and procedures in three main maternity facilities in the capital, Dili to examine the extent to which the WHO (2016) standards for women's experiences of care have been adopted.

### Results

Despite the availability of global guidelines, key standards to improve women's experience of care have not been included in the National Intrapartum Care guidelines in Timor-Leste. There was no mention of avoiding mistreatment of women, needing informed consent for procedures, or strengthening women's own capability and confidence. In the policy wording, women tended to be distanced from the care 'procedures' and the protocols could be improved by taking a more woman-centred approach. The results of the health facility assessment showed extremely low use of standards that improve women's experiences of care. Health Facility 1 and 2 met two of the 21 quality measures, while Health Facility 3 met none of them.

### Conclusion

The discourse communicated through policy fundamentally affects how health care issues are framed and how policies are enacted. Given the findings of this study, combined with

**Funding:** This research was funded through AF's PhD scholarship by Airlangga University, Indonesia and Instituto Superior Cristal, Timor-Leste. The funders had no role in study design, data collection and analysis, decision to publish, or preparation of the manuscript.

previously documented issues around quality of care and low satisfaction with maternal health services, there is a need for a fundamental shift in the culture of care for women. This will require an immediate focus on leadership, training and policy-frameworks to increase respectful care for women in health facilities. It will also require longer-term effort to address the power imbalances that drive mistreatment of women within and across social systems, and to support models of care that inherently foster understanding and compassion.

## Introduction

The care women receive during pregnancy, birth and in the postnatal period is intimately linked to their well-being and that of their newborn infants. Poor quality of maternal health care across the perinatal period is associated with increased risk of birth trauma [1, 2], birth and postpartum complications [3] and has flow-on effects for maternal and infant mortality [4–6]. Improving the quality of maternal healthcare is integral to women's and children's right to health, contributing to their equitable access and positive experience with health care [7].

Respectful maternity care is care that is organised and provided to women in a way that maintains their dignity, privacy and confidentiality, ensures freedom from harm and mistreatment, and enables informed choice and continuous support during labour and birth [7]. The WHO [5] framework for quality maternal and newborn health care establishes measurable standards of care that provide important guidance for policy-makers, managers and researchers, to assess strengths and gaps in health service-systems and develop processes to improve care over time. The framework includes eight standards covering:

### Provision of care

1. Evidence based practices for routine care and management of complications

2. Actionable information systems

3. Functional referral systems

### Experience of care

4. Effective communication

5. Respect and preservation of dignity

6. Emotional support

### Cross-cutting standards

7. Competent, motivated human resources

8. Essential physical resources available

Importantly, the framework incorporates women's experience of care, including standards for effective communication; respect and preservation of dignity; and emotional support (Standards 4–6). When issues central to women's experience of care are left out of health policy and facility protocols, women and families are more likely to be exposed to disrespectful, abusive

and coercive behaviour by healthcare providers [8]. This can culminate in a highly traumatic experience of birth [1], feelings such as fear of childbirth [9] and avoidance of healthcare [10].

The woman-centred care agenda, foregrounding women's experiences of care, has gained traction globally and national health systems are starting to adopt these global standards. Despite this progress, gains can be slower in poorly resourced health systems [11]. In Timor-Leste, the quality of maternal health care has long been identified as a central health systems issue [12–15]. While maternal mortality has steadily declined since the country gained independence from Indonesia in 2002, the 2016 Demographic and Health Survey (DHS) showed only 48% of women gave birth in a health facility, and the steepest decline was between first birth and second birth (nationally, 63% of first births occurred in a health facility, 48% for second or third birth, 41% for fourth or fifth birth, and 35% for sixth or more) [16]. Women's birth choices are complex [15] and the contribution of poor quality of care toward preference for home birth has not yet been quantified. However poor quality of care is likely to be a factor in low utilisation of health services for birth as the national DHS showed 56% of women had concerns about being treated respectfully when accessing health care [16].

Given such prevalence of concern raised by women, and the importance of clear policies and frameworks to start addressing the issue at multiple levels [17], we examine the adoption of the WHO [5] Standards for Improving Quality of Maternal and Newborn Care in Timor-Leste. The study uses the WHO Experience of Care Standards as a framework to review the national Timor-Leste Intrapartum Care Guidelines [18, 19] and perform an audit of site-specific policies and protocols at three major health facilities in the country's capital, Dili.

## Methods

### Study design

For this mixed methods study we adopted a critical perspective of policy transfer [20]. The research consisted of a qualitative policy review and a quantitative health facility audit, based on the WHO [5] Standards for Improving Quality of Maternal and Newborn Care in Health Facilities. This study was part of larger program of research being undertaken by author AF, on women's experiences of disrespect and abuse during childbirth in Dili (initial qualitative study, survey with women on prevalence of mistreatment, and this policy assessment). In this way, the findings presented here are part of a broader examination of the provision and experience of care for women giving birth in Dili. The aim is to inform gaps and ways forward to achieve high quality, respectful maternity care in Timor-Leste.

### Sampling technique

The study was conducted in the capital city, Dili, as this municipality has the highest proportion of facility-based births (83% of births in a health facility, compared with 48% nationally) [16]. Dili also contains the largest referral hospital in the country as well as six community health centres providing primary health care and birthing services. Three health facilities were sampled based on the highest number of births per month. The facilities have been de-identified and are denoted as Health Facility 1, 2 and 3.

### Ethical approval and permission

Ethical clearance was obtained from the Ethical Review Committee at Airlangga University, Indonesia (ID# 071/HRECC.FODM/III/2022). Reciprocal approval was obtained from Timor-Leste's Human Ethics Committee at the National Institute for Health (INS–*Instituto Nasional*

*Saude*) in Dili (ID# 603MS-INS/GDE/IV/2022). At each of the study sites, the health facility or maternity department manager was contacted, the research explained and a letter of approval from INS was provided. The study was also approved through the ethics department of the larger study facility.

## Policy review

Deductive content analysis was used to compare the extent to which the concepts in the WHO [5] Standards and Quality Statements (4–6 on experiences of care) had been incorporated into the Timor-Leste National Intrapartum Standards of Care and Clinical Protocols [18, 19]. The national policies were first read and reviewed as a whole, then a word search was performed to further identify specific concepts and how they were framed within the documents.

## Health facility audit

An audit tool (Tables 2–7, S1 Table) was derived from the WHO [5] Standards 4–6 on Experiences of Care. These included the Input Quality Measures for:

- Standard 4 on Effective Communication: 4.1 All women and their families receive information about the care and have effective interactions with staff.

- Standard 5 on Respect and Preservation of Dignity: 5.1 All women and newborns have privacy around the time of labour and childbirth, and their confidentiality is respected; 5.2 No woman or newborn is subjected to mistreatment, such as physical, sexual or verbal abuse, discrimination, neglect, detainment, extortion or denial of services; 5.3 All women can make informed choices about the services they receive, and the reasons for interventions or outcomes are clearly explained.

- Standard 6 on Emotional Support: 6.1 Every woman is offered the option to experience labour and childbirth with the companion of her choice; 6.2 Every woman receives support to strengthen her capability during childbirth.

The 'Input Quality Measures' for each of these standards was developed into a table, with a tick-box column to indicate whether the criteria were present or not at each facility (Tables 2–7, S1 Table). The wording for each quality measure was adapted for the context of Timor-Leste through rephrasing for clarity, and then translated to Indonesian as the research was being undertaken at an Indonesian University. For example, WHO Standard 4.1 '*Easily understood health education materials, in an accessible written or pictorial format, are available in the languages of the communities served by the health facility*' was rephrased as '*Health education materials are available in Tetum, with easily accessible writing or pictures, and are given by midwives to birthing women*'. Some input measures were omitted if they were not applicable to the Timor-Leste context, for example standards referring to detainment for non-payment, as Timor-Leste has a free public healthcare system.

At each study facility a department manager was contacted, the research was explained and a time was made to visit the service. Interviews were conducted with maternity department managers from May to July 2022. Written informed consent was provided by the participant and they were interviewed by AF in the local language, Tetun, to assess whether each of the input criteria were met. Item responses consisted of "Yes", meaning the health facility had the policy, guideline or equipment in place, or "No" meaning it did not exist at the time of the interview. Where policy was only somewhat compliant with the standard, participants were probed for further comment until consensus was reached.

## Limitations

This study sampled three of the seven health facilities with birthing services in the capital city Dili, and is not representative of all services. Although only urban health facilities were sampled, the findings are not likely to be more positive in rural facilities given the time it takes policies to trickle down. Several of the standards relate to whether regular training is provided. However, training has been disrupted due to COVID-19, particularly over the two years preceding the survey. It is possible that staff may have attended broader training that incorporates aspects of respectful care, for example Emergency Obstetric Care training. There also could have been policies and guidelines that the person interviewed did not know about. However, the heads of the maternity units usually participate in training and coordination meetings and were the most well-positioned to know about policies and protocols at their facility.

## Results

### Adoption of WHO standards in national intrapartum care policies

The review of the Timor-Leste national intrapartum Standards of Care and Clinical Protocols for Community Health Centres [18] and Referral Facilities [19] showed that the WHO Experience of Care Standards were only marginally adopted. The Timor-Leste guidelines briefly referenced supportive care, in the form of the following, on page 113: [18, 19]:

Supportive care

- Respect for clients (respect during care and discussions and maintain privacy during examinations)

- Communication with patient and family about progress and problems and management.

- Maintain cleanliness of patient by encouraging client to shower, clean genitalia

- Encourage the patient to walk around

- Encourage to pass urine

- Make sure patient is adequately hydrated and nourished by drinking adequate fluids and eating light meals

- Pain and discomfort relief as needed

- If requested, allow one accompanying person of women's choice in labour room.

When mapped to the WHO standards, the only reference to effective communication was one dot point stating 'Communication with patient and family about progress and problems and management'. For standard 5 on respect and preservation of dignity, there was one dot point in the document stating 'Respect for clients (respect during care and discussions and maintain privacy during examinations)'. Within this concept of respect, there was no mention of not mistreating women or newborns, and no mention of avoiding physical, sexual or verbal abuse, discrimination, neglect, detainment, extortion or denial of services. Similarly, there was no mention of informed choice, consent or that reasons for interventions or outcomes should be clearly explained. For standard 6, emotional support, there was one dot point stating '**If requested**, allow one accompanying person of women's choice in labour room' (author emphasis added). There were no statements around strengthening a woman's own capability and confidence during childbirth. Overall, any reference to supportive care in the national protocols was very sparse; they tended to be general statements, without specific actions or directions on how to achieve them (i.e. Comfort: physical and emotional). In future policy wording,

it is important to acknowledge women as active agents in the birth process, and provide specific examples of how practitioners can support women's social and emotional wellbeing and their right to informed choice.

## Health facility assessment

The findings of the health facility assessment show all of the sampled facilities scored low on quality measures for Experiences of Care (Table 1). Health Facility 1 and 2 had the highest score, meeting two of the 21 criteria, while Health Facility 3 met none of them. Following is a breakdown of challenges within each domain.

**Effective communication.** None of the WHO recommended standards for supporting effective communication between women and their care providers were in place in any of the surveyed health facilities (Table 2). There were no birth education materials for women available at the facilities (for example on pain management, relaxation, how to push during labour), and there were no specific policies in place to promote midwives' communication and counselling skills. One of the health facilities had received in-service training on interpersonal communication (provided by INS). However, due to COVID-19 it was not provided regularly and they did not receive follow-up training or support. The main communication issues identified by the interview participants were midwives not introducing themselves to women, not involving women in choices about what actions were taken during their labour and birth and not providing opportunities to discuss women's concerns. There were also no induction or supervision mechanisms to support midwives' communication and counselling skills.

**Privacy and confidentiality.** Health facility 1 and 2 had a separate room for birth, which was small (3 x 4 meters). There were sometimes difficulties fitting people in the room, especially when students were observing. The third health facility had two beds in the 'delivery'

**Table 1. Number (%) of experience of care quality measures met at each study facility.**

| Experience of care quality measures | Number (%) of quality measures met | | |
|---|---|---|---|
| | **Health Facility 1** | **Health Facility 2** | **Health Facility 3** |
| Effective communication | 0/4 (0%) | 0/4 (0%) | 0/4 (0%) |
| Privacy and confidentiality | 1/2 (50%) | 1/2 (50%) | 0/2 (0%) |
| Avoid mistreatment | 0/5 (0%) | 0/5 (0%) | 0/5 (0%) |
| Informed consent | 0/3 (0%) | 0/3 (0%) | 0/3 (0%) |
| Companion of choice | 1/4 (25%) | 1/4 (25%) | 0/4 (0%) |
| Strengthen women's capability | 0/3 (0%) | 0/3 (0%) | 0/3 (0%) |
| Total | 2/21 (10%) | 2/21 (10%) | 0/21 (0%) |

**Table 2. Quality measures for effective communication.**

| No | Quality measures | Health facility 1 | Health facility 2 | Health facility 3 |
|---|---|---|---|---|
| 1 | Health education materials are available in Tetum, with easily accessible writing or pictures, and are given by midwives to birthing women. | No | No | No |
| 2 | Midwives are oriented and receive in-service training at least once every 12 months to improve communication and interpersonal counselling skills according to local culture. | No | No | No |
| 3 | The maternity department has a written, up-to-date policy that outlines clear goals, operational plans and monitoring mechanisms to promote the interpersonal communication and counselling skills of midwives. | No | No | No |
| 4 | Midwives in the maternity unit receive supportive supervision in interpersonal communication, counselling and cultural competence every three months. | No | No | No |

room. This was a problem when two women were giving birth, and the companion of the other birthing woman could either see and overhear, or would be asked to leave the room. There was also lack of space and privacy for women in labour, who often had to stay on the postnatal ward (large shared room) due to lack of space. When curtains and partitions were present, they were not always used by the health providers, and curtains provide no sound protection. In addition, there were no written protocols to guide staff on the importance of or how to promote privacy and confidentiality (Table 3).

**Avoid mistreatment.** There were no policies outlining the need to avoid mistreatment of women, accountability mechanisms, or the rights of women to make complaints (Table 4). Although there was a complaints box at one of the health facilities, there was no mechanism to check or review it, and it was empty during the time of the interview. Participants at all facilities said their midwifery staff had not received in-service training or supportive supervision about avoiding mistreatment or respecting women's rights.

**Informed consent.** There were no written policies outlining the need to obtain women's informed consent before examinations or procedures such as episiotomy, vaginal examination, or perineal suturing (Table 5). While there was a consent procedure to perform cesarean section, there were no standard consent forms available and a lack of process around seeking

**Table 3. Quality measures for privacy and confidentiality.**

| No | Quality measures | Health facility 1 | Health facility 2 | Health facility 3 |
|---|---|---|---|---|
| 1 | The physical environment of the health facility allows for privacy and respectful, confidential care, including the availability of curtains, screens, partitions and adequate bed capacity. | Yes | Yes | No |
| 2 | The health facility has written and up-to-date protocols to ensure privacy and confidentiality for all women during birth. | No | No | No |

**Table 4. Quality measures to ensure no woman is subjected to mistreatment.**

| No | Quality measures | Health facility 1 | Health facility 2 | Health facility 3 |
|---|---|---|---|---|
| 1 | The health facility has written, up-to-date, zero-tolerance non-discriminatory policies with regard to the mistreatment of women in the maternity ward. | No | No | No |
| 2 | The health facility has a written accountability mechanism in the event of mistreatment or violence | No | No | No |
| 3 | The health facility has up-to-date written policies and protocols outlining the rights of women to make complaints about the care received and has an easily accessible mechanism (e.g. comments box) for submitting complaints. | No | No | No |
| 4 | Midwives receive in-service training and supportive supervision in respecting the rights of women, and providing respectful care. Orientation is given to new staff. | No | No | No |
| 5 | The health facility has a complaints box, which is easily accessible to women and their families, which is periodically emptied and the contents reviewed. | No | No | No |

**Table 5. Quality measures for informed choice.**

| No | Quality measures | Health facility 1 | Health facility 2 | Health facility 3 |
|---|---|---|---|---|
| 1 | The health facility has a written and up-to-date policy for obtaining the consent of the woman prior to examinations and procedures. | No | No | No |
| 2 | Health facilities have standard informed consent forms that help midwives provide women with easy-to-understand information to obtain full consent before taking action | No | No | No |
| 3 | Midwives in the maternity unit receive in-service training and supportive supervision in informed consent and women's right to choose their care. Orientation is provided for new staff. | No | No | No |

**Table 6. Quality measures for companion of choice.**

| No | Quality measures | Health facility 1 | Health facility 2 | Health facility 3 |
|---|---|---|---|---|
| 1 | The labour and childbirth areas are organized in such a way as to allow a physical private space for the woman and her companion at the time of birth. | Yes | Yes | No |
| 2 | The health facility has a written, up-to-date protocol, which is explained to women and their families, to encourage all women to have at least one person of their choice, as culturally appropriate, with them during labour, childbirth and the immediate postnatal period. | No | No | No |
| 3 | Midwives in the health facility are oriented and receive in-service refresher training sessions at least once every 12 months on the evidence for and positive impact of the presence of a chosen companion during labour and birth. | No | No | No |
| 4 | Orientation sessions and information (written or pictorial) are available to orient the companion on his or her role in supporting the woman during labour and birth. | No | No | No |

**Table 7. Quality measures to strengthen women's capability.**

| No | Quality measures | Health facility 1 | Health facility 2 | Health facility 3 |
|---|---|---|---|---|
| 1 | Midwives in the labour and childbirth areas of the maternity unity were oriented in nonpharmacological and pharmacological pain relief and received in-service training or sessions at least once in the preceding 12 months. | No | No | No |
| 2 | The health facility has a written, up-to-date protocol, which is explained to women and their families, to minimise unnecessary interventions, support normal labour and strengthen the woman's capability, so that she feels in control of her childbirth experience. | No | No | No |
| 3 | Midwives in the labour and childbirth areas of the maternity unit were oriented and received in-service training or refresher sessions at least once in the preceding 12 months to strengthen their interpersonal and cultural competence in providing emotional support. | No | No | No |

consent for other procedures such as vaginal examination and episiotomy. In addition, participants reported that their midwifery staff had not received in-service training or supportive supervision on informed consent procedures or women's right to choose their care.

**Companion of choice.** There were no written protocols that encouraged women to have a companion of choice during labour, birth or post-partum (Table 6). One participant said their staff often asked women if they would like a family member to accompany them. The shared birthing room in one of the health facilities was particularly problematic for male birth companions, who were not allowed to enter the room when there was more than one woman using the space. Midwives in the study facilities had not been provided with in-service training on the positive impact of labour companion, and there was no information to orient companions on their role in supporting women during birth.

**Strengthen women's capability.** Interview participants said that their midwifery staff had not been oriented in the use of natural methods of pain relief, nor in strengthening their ability to provide emotional support, and had not received in-service training in relation to these. There were no protocols to minimise unnecessary interventions or strengthen a woman's own capability during birth (Table 7).

## Discussion

Despite availability of global guidelines to support respectful maternity care and prevent mistreatment of women during childbirth [5, 7, 21, 22], key standards to improve women's experience of care do not appear in the national intrapartum care guidelines in Timor-Leste [18, 19]. Most significantly, there was no mention of avoiding mistreatment of women, needing informed consent for procedures, or strengthening women's own capability and confidence.

Given the dearth of national policies and guidelines that prioritise women's experience of care, it is not surprising that the audited health facilities met so few of the Experience of Care standards. Health Facility 1 and 2 had the highest score, meeting two of the 21 quality measures, while Health Facility 3 met none of them. Without policies that prioritise respectful woman-centred care, women are at risk of disrespect directly as a result of harmful policies or protocols [23], or indirectly as a result of lack of guidance, oversight and procedural consequence for neglectful or harmful practice [11, 24].

Qualitative research on women's experience of mistreatment, conducted by the authors in the same audited health facilities, revealed that women were subjected to frequent physical and verbal abuse by their birth attendant [25]. This included hitting her, forcing her thighs open, stitching her perineum without pain relief, shouting and being angry, and humiliating her. The abuse was most severe in the second stage of labour, during the intense pain of birth, and when the birth attendant perceived the woman was not following their instructions [25].

The discourse communicated through policy, the tone and wording within documents, fundamentally affects how health care issues are framed and how policies are enacted [26]. In the national intra-partum care guidelines [18, 19] women were distanced from the care 'procedures', their views and wishes were rarely mentioned. Rather, the wording centred around guiding and directing women during the birth process. The lack of attention given to women's experiences of care in maternal health policy and in health facilities, mirrors broader gender inequality in Timorese society [27]. Given the high rates of family violence in Timor-Leste (35% experienced violence from their partner in the past 12 months), in combination with the high fertility rate of 4.2 births per woman [16], there is likely to be substantial intersections of abuse and trauma for women, repeated in multiple settings and over multiple births.

Policy improvements are fundamental to practice improvements, as adequate policies, procedures and accountability mechanisms are linked to behaviour of health care providers, with flow on effects for improved birth experiences and better outcomes for women and families [8]. In addition to policy guidance, calls for improvements in training and supportive supervision are echoed across existing literature [2, 8, 21, 28–31]. Training should include confronting the normalisation of violence against women within families and across systems, the belief that mistreatment is necessary to minimise clinical harm, reflection on our own psychological and emotional responses in various scenarios [32] and strategies that encouraged introspection and cultural humility [33].

There is an immediate need for greater awareness, training and policy frameworks in Timor-Leste that directly prevent mistreatment of women during birth. Evidence suggests, however, that simply having policies and training is not enough, as health systems interact with structural inequality and individual interpersonal dynamics to underpin the disrespect and abuse of women [34]. Key organisational factors such as high workload, low pay, resource shortages, and lack of professional autonomy and supervision and feedback mechanisms have been identified as contributing factors that need to be addressed at a systems level [35]. The WHO [21] has long argued that ending disrespect and abuse during childbirth must be approached comprehensively. This has been reinforced in recent reviews of strategies to prevent mistreatment, in that they emphasise the need to address the power-related drivers of mistreatment across the ecological model (i.e. intrapersonal, interpersonal, community, organizational, and law/policy) [36], as well as transforming institutional leadership and approaches to supervision that tackle existing power hierarchies [35]. Given the findings of this study, combined with the widespread documentation of poor quality of care and low satisfaction with maternal health services [12–16, 25, 37], there is a need for urgent support to shift the culture of care for women. This shift needs to be based on the diverse experiences of

birthing women in Timor-Leste, and their vision of what better models of maternity care look like in the future.

## Conclusion

Lack of policy frameworks that prioritise women's experiences of care hinders the ability to implement respectful ways of working with women in maternal health services. National intrapartum care guidelines could be improved by incorporating women's experiences of care, paying attention to women's right to choice and freedom from violence. Further work is required to understand how to best support the respectful maternity care agenda in Timor-Leste, and what models of care inherently foster understanding and compassion in health services.

## Supporting information

**S1 Table. Quality measures for experience of care.**
(DOCX)

## Acknowledgments

Thank you to the research participants and department heads who gave their permission to participate in this study. Thanks to GdA for review and input into a draft of this manuscript.

## Author Contributions

**Conceptualization:** Angelina da Costa Fernandes, Stefanus Supriyanto, Chatarina Umbul Wahyuni, Hari Basuki Notobroto.

**Data curation:** Angelina da Costa Fernandes, Alexandra Gregory, Kayli Wild.

**Formal analysis:** Angelina da Costa Fernandes, Alexandra Gregory, Kayli Wild.

**Funding acquisition:** Angelina da Costa Fernandes.

**Investigation:** Angelina da Costa Fernandes.

**Methodology:** Angelina da Costa Fernandes, Stefanus Supriyanto, Chatarina Umbul Wahyuni, Hari Basuki Notobroto, Alexandra Gregory, Kayli Wild.

**Project administration:** Angelina da Costa Fernandes.

**Supervision:** Stefanus Supriyanto, Chatarina Umbul Wahyuni, Hari Basuki Notobroto, Kayli Wild.

**Writing – original draft:** Angelina da Costa Fernandes, Alexandra Gregory, Kayli Wild.

**Writing – review & editing:** Angelina da Costa Fernandes, Stefanus Supriyanto, Chatarina Umbul Wahyuni, Hari Basuki Notobroto.

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
