## [Decision Letter · Decision Letter 0]

8 Nov 2023

PONE-D-23-21975Adoption of policies to improve respectful maternity care in Timor-LestePLOS ONE

Dear Dr. Fernandes,

Thank you for submitting your manuscript to PLOS ONE. After careful consideration, we feel that it has merit but does not fully meet PLOS ONE’s publication criteria as it currently stands. Therefore, we invite you to submit a revised version of the manuscript that addresses the points raised during the review process.

ACADEMIC EDITOR:

Dear authors, 

Please go through the article and comply the points raised by the reviewers before assessing for suitability of publication.

With regards,

Ranjit

We look forward to receiving your revised manuscript.

Kind regards,

Ranjit Kumar Dehury

Academic Editor

PLOS ONE

Journal Requirements:

"This research was funded through AF’s PhD scholarship by Airlangga University, Indonesia and Instituto Superior Cristal, Timor-Leste"

"This research was funded through AF’s PhD scholarship by Airlangga University, Indonesia and Instituto Superior Cristal, Timor-Leste. "

"This research was funded through AF’s PhD scholarship by Airlangga University, Indonesia and Instituto Superior Cristal, Timor-Leste. "       

5. Please include captions for your Supporting Information files at the end of your manuscript, and update any in-text citations to match accordingly. Please see our Supporting Information guidelines for more information: http://journals.plos.org/plosone/s/supporting-information

Additional Editor Comments:

Dear authors,

Please go through the article and comply the points raised by the reviewers before assessing for suitability of publication.

With regards,

Ranjit

Reviewers' comments:

Reviewer's Responses to Questions

**Comments to the Author**

1. Is the manuscript technically sound, and do the data support the conclusions?

Reviewer #1: Yes

Reviewer #2: Yes

2. Has the statistical analysis been performed appropriately and rigorously? 

Reviewer #1: Yes

Reviewer #2: Yes

3. Have the authors made all data underlying the findings in their manuscript fully available?

Reviewer #1: Yes

Reviewer #2: Yes

4. Is the manuscript presented in an intelligible fashion and written in standard English?

Reviewer #1: Yes

Reviewer #2: Yes

5. Review Comments to the Author

Reviewer #1: • WHO standards and quality statements (4-6 on experience of care) need elaboration. For example, what is 4-6 on experience of care? Briefly introduce the WHO standards from one to N and why the 4-6 is chosen. The author/s must introduce the standards appropriately.

• The questions pertaining to ‘input quality measures’ at the facilities were recorded in a tick box column; ‘yes’ or ‘no’ type. The Likert scale method could have been more appropriate.

• What are the “four dot points” in the Timor-Leste guidelines?

• In the introduction, with the use of data, it is shown that non-institutional deliveries increase with second and third childbirths; was this phenomenon explored through the qualitative interviews? If yes, what were the gender or power dynamics in that behaviour?

Reviewer #2: The overall paper underlines the importance of policy discourse and its influence on healthcare practices. It rightly calls for a fundamental shift in the culture of care for women in Timor-Leste and suggests a two-pronged approach involving immediate actions and long-term strategies. This paper provides a logical and actionable way forward based on the study's findings.

The paper does not mention any potential limitations of the study, such as the scope, sample size, or potential biases. Including this information would offer a more complete understanding of the research's boundaries. Additionally, it does not explicitly state whether this is a qualitative or quantitative study, which could be important for readers seeking specific research methods.

6. PLOS authors have the option to publish the peer review history of their article (what does this mean?). If published, this will include your full peer review and any attached files.

Reviewer #1: No

Reviewer #2: **Yes: **Abhishek Dondapati

---

## [Author Response · Author response to Decision Letter 0]

30 Dec 2023

Journal requirements

Response: We have re-named the files according to the guidelines.

2. We note that you have provided additional information within the Acknowledgements Section that is not currently declared in your Funding Statement. Please note that funding information should not appear in the Acknowledgments section or other areas of your manuscript. We will only publish funding information present in the Funding Statement section of the online submission form. Please remove any funding-related text from the manuscript and let us know how you would like to update your Funding Statement. 

Response: The following sentence has been removed from the acknowledgements section in the manuscript “This research was funded through AF’s PhD scholarship by Airlangga University, Indonesia and Instituto Superior Cristal, Timor-Leste.

3. Please state what role the funders took in the study. If the funders had no role, please state: "The funders had no role in study design, data collection and analysis, decision to publish, or preparation of the manuscript." Please include this amended Role of Funder statement in your cover letter; we will change the online submission form on your behalf.

Response: The following sentence should be added to the online submission “This research was funded through AF’s PhD scholarship by Airlangga University, Indonesia and Instituto Superior Cristal, Timor-Leste. The funders had no role in study design, data collection and analysis, decision to publish, or preparation of the manuscript.” 

4. Upon re-submitting your revised manuscript, please upload your study’s minimal underlying data set as either Supporting Information files or to a stable, public repository and include the relevant URLs, DOIs, or accession numbers within your revised cover letter.

Response: The raw data tables are already presented within the manuscript. We have also attached as a separate file under supporting information. 

5. Please include captions for your Supporting Information files at the end of your manuscript, and update any in-text citations to match accordingly. Please see our Supporting Information guidelines for more information: http://journals.plos.org/plosone/s/supporting-information

Response: We have included captions for the supporting files and made reference to them within the text to match (Line 127 and 137). 

6. Please review your reference list to ensure that it is complete and correct. 

Response: We have reviewed and amended the references list. 

Reviewer #1: 

• WHO standards and quality statements (4-6 on experience of care) need elaboration. For example, what is 4-6 on experience of care? Briefly introduce the WHO standards from one to N and why the 4-6 is chosen. The author/s must introduce the standards appropriately.

Response: We have outlined in more detail the 8 standards in the WHO framework, and linked it more clearly to the importance of focusing on the domain of ‘experiences of care’. See lines 56-75. 

• The questions pertaining to ‘input quality measures’ at the facilities were recorded in a tick box column; ‘yes’ or ‘no’ type. The Likert scale method could have been more appropriate.

Response: The guidelines, protocols and equipment are either present or absent within the health facilities, therefore the appropriate measure is yes/no, not a continuous variable. 

• What are the “four dot points” in the Timor-Leste guidelines?

Response: We have clarified the dot points as they appear in the Timor-Leste guidelines, and further illustrated how they relate to the WHO standards of care (line 166-187). 

• In the introduction, with the use of data, it is shown that non-institutional deliveries increase with second and third childbirths; was this phenomenon explored through the qualitative interviews? If yes, what were the gender or power dynamics in that behaviour?

Response: This research was designed as an assessment of policy, guidelines and procedures. The sociological dynamics of where women give birth and the power dynamics surrounding that are beyond the scope of this paper. We have, however, referenced a paper that has been published on women’s birth choices in Timor-Leste and the individual, family and social dimensions that influence place of birth (Line 83). 

Reviewer #2: 

The overall paper underlines the importance of policy discourse and its influence on healthcare practices. It rightly calls for a fundamental shift in the culture of care for women in Timor-Leste and suggests a two-pronged approach involving immediate actions and long-term strategies. This paper provides a logical and actionable way forward based on the study's findings.

Response: Thank you for acknowledging the importance of this work. 

The paper does not mention any potential limitations of the study, such as the scope, sample size, or potential biases. Including this information would offer a more complete understanding of the research's boundaries. 

Response: Limitations have been added at the end of the methods (Line 153-159). 

Additionally, it does not explicitly state whether this is a qualitative or quantitative study, which could be important for readers seeking specific research methods.

Response: We have clarified that this is a mixed methods study consisting of qualitative review of national policy (deductive content analysis) and quantitative survey of health facilities (Line 96-97).

---

## [Decision Letter · Decision Letter 1]

23 Jan 2024

Adoption of policies to improve respectful maternity care in Timor-Leste

PONE-D-23-21975R1

Dear Dr. Fernandes,

We’re pleased to inform you that your manuscript has been judged scientifically suitable for publication and will be formally accepted for publication once it meets all outstanding technical requirements.

Kind regards,

Ranjit Kumar Dehury

Academic Editor

PLOS ONE

Additional Editor Comments (optional):

Reviewers' comments:

Reviewer's Responses to Questions

**Comments to the Author**

1. If the authors have adequately addressed your comments raised in a previous round of review and you feel that this manuscript is now acceptable for publication, you may indicate that here to bypass the “Comments to the Author” section, enter your conflict of interest statement in the “Confidential to Editor” section, and submit your "Accept" recommendation.

Reviewer #1: All comments have been addressed

Reviewer #2: All comments have been addressed

2. Is the manuscript technically sound, and do the data support the conclusions?

Reviewer #1: Yes

Reviewer #2: Yes

3. Has the statistical analysis been performed appropriately and rigorously? 

Reviewer #1: Yes

Reviewer #2: Yes

4. Have the authors made all data underlying the findings in their manuscript fully available?

Reviewer #1: Yes

Reviewer #2: Yes

5. Is the manuscript presented in an intelligible fashion and written in standard English?

Reviewer #1: Yes

Reviewer #2: Yes

6. Review Comments to the Author

Reviewer #1: This manuscript is now acceptable for publication. Statistical tools are applied appropriately to draw the conclusion. The authors have made the data accessible used in the research. The standard of the language are appropriately taken care of. Authors should be conscious about the dual publication of the same research.

Reviewer #2: (No Response)

7. PLOS authors have the option to publish the peer review history of their article (what does this mean?). If published, this will include your full peer review and any attached files.

Reviewer #1: **Yes: **Imteyaz Ahmad

Reviewer #2: **Yes: **Abhishek Dondapati

---

## [Editor Report · Acceptance letter]

4 Mar 2024

PONE-D-23-21975R1 

PLOS ONE

Dear Dr. Fernandes, 

I'm pleased to inform you that your manuscript has been deemed suitable for publication in PLOS ONE. Congratulations! Your manuscript is now being handed over to our production team.

Kind regards, 

on behalf of

Dr. Ranjit Kumar Dehury 

Academic Editor

PLOS ONE